# Sewage Sludge Fertilization—A Case Study of Sweet Potato Yield and Heavy Metal Accumulation

Carla Ragonezi [1,2,*], Nuno Nunes [1,2], Maria Cristina O. Oliveira [1], José G. R. de Freitas [1], José Filipe T. Gançança [1] and Miguel Â. A. Pinheiro de Carvalho [1,2,3]

[1] ISOPlexis Centre Sustainable Agriculture and Food Technology, Campus da Penteada, University of Madeira, 9020-105 Funchal, Portugal

[2] Centre for the Research and Technology of Agro-Environmental and Biological Sciences (CITAB), University of Trás-os-Montes and Alto Douro, 5000-801 Vila Real, Portugal

[3] Faculty of Life Sciences, Campus da Penteada, University of Madeira, 9020-105 Funchal, Portugal

* Correspondence: carla.ragonezi@staff.uma.pt; Tel.: +351-291-705-002

**Abstract:** Sewage sludge (SS) is derived from wastewater treatment plants and can be used as a biofertilizer when properly stabilized. This work aimed to evaluate SS application for agricultural production improvement. SS was tested on Porto Santo Island (Portugal). The experiment was randomly designed with three 25 $m^2$ plots for each treatment (2 SS concentrations + control without SS) and performed in two consecutive cycles. For the first cycle, dehydrated sludge was mixed with soil, obtaining final concentrations of 0.8 $kg/m^2$ (C1) and 1.6 $kg/m^2$ (C2). Half of the concentration was used for the second cycle. Fifty-eight sweet potato plants were used in each plot. SS application boosted the agronomic parameters of biomass, productivity, and shoot biomass. Furthermore, improvements in soil properties were observed, mainly for pH, CEC, and $NO_3$-N, with no significant increase in heavy metals. For the edible parts, heavy metal concentrations decreased, and Pb was the only one that still exceeded the maximum limits. The results demonstrated that SS application to low-fertility soil is effective in improving the agronomic parameters of sweet potato and enhancing soil features. Further studies considering other variables, i.e., SS origin, soil properties, and the crop, must be carried out to propose custom applications.

**Keywords:** agronomic parameters; production; soil properties; soil amendment; bioaccumulation; translocation factors





## 1. Introduction

Sewage sludge (SS) is a by-product of wastewater treatment plants (WWTP); it is a semi-solid waste derived from domestic and industrial effluents and subjected to physical, chemical, and biological treatments, which generally include three stages, including a second treatment that modifies this waste into a biosolid [1,2]. It can be used as an organic fertilizer and soil conditioner in agriculture since it improves soil nutrient properties (NPK and micronutrients) as well as physical characteristics (increases water retention, lowers bulk density, and improves cation exchange capabilities) [3,4]. Along with these tempting attributes, SS also contains contaminants, including hormones, pathogens, pharmaceutical chemicals, and heavy metals that can imprint a detrimental effect on crop production and cross over to human beings through the food chain [5]. Several techniques and strategies were developed to reduce this risk, including SS treatment and proper agronomic application of this biomass. Thermal treatments for the stabilization of heavy metals include incineration, pyrolysis, gasification, hydrothermal carbonization, and supercritical water gasification, which are treatments that occur after a preliminary dehydration step to reduce biomass volume and improve treatment effectiveness [5]. Aerobic and anaerobic digestion of SS are described in the literature as efficient methods to reduce the degree of this biomass contamination and increase economic impact. For example, the stabilization

mechanism of SS was studied by employing a one-stage, aerobic thermophilic digestion, removing volatile solids at 55 °C up to 45% in 552 h [6]. Furthermore, the addition of ferric nitrate to aerobic digestion developed significant removal of volatile solids, shortening 7 days of stabilization time at 55 °C [7]. Additionally, combining aerobic digestion with bioleaching, which uses microorganisms to reduce the system pH, efficiently removes heavy metals and increases dewatering ability, maintaining the agricultural potential of the biomass [8]. Martín et al. determined that 19 of 22 pharmaceutically active compounds are present in SS, and anaerobic digestion considerably reduces this content [9]. Several works complemented the anaerobic digestion with additional treatments such as continuously stirred tank reactors under thermophilic (50 °C) and mesophilic (37 °C) conditions [10] and focused-pulsed treatment, a vanguard technology introduced to reduce operating costs, increase methane production, and reduce biosolids [11]. Other methods of reducing the hazard risk of contamination include adding fly ash and/or lime [12], mixing with steelmaking slag [13], composting [14], and vermicomposting [15].

Each EU country may favor different options to manage the SS produced, for example, in agricultural use (directly or after composting), incineration, landfills, energy production, or others. In Portugal, the main option is agricultural valorization, which represents more than 50% of the SS produced [16], but for the Madeira archipelago, this option has just begun to be explored. The WWTP located on the island of Porto Santo was installed to cope with a maximum flow of 4000 m$^3$/day and produces 265 tons of dried biosolids annually, treating all of the island's domestic wastewater with the final treated and disinfected effluent used for agricultural irrigation [17]. The dried biosolids were envisioned, in this study, to be used as a fertilizer, according to the EU directive 86/278/EEC [18] and FAO [19], also taking into consideration the EC No. 1881/2006 of 19 December 2006 [20], determining the maximum permissible levels of food contaminants in foodstuffs, and keeping the toxicological levels acceptable. There is increasing research regarding the utilization of SS as a fertilizer, and liquid fertilizers (through alkaline thermal hydrolysis) intended for use as nitrogen-rich plant-growth-promoting nutrients and biostimulants are already being developed [21]. Field trials using this biomass to assess its effect on crop production are frequent, including crops such as French beans [22], mung beans [23], wheat [24], beets [25], barley, and Chinese cabbage [26]. Additionally, the focus is being given to the optimization of organomineral fertilizers for maize, sunflower [27], and rapeseed crops [28]. From an energy security perspective, SS is being integrated into energy crop field trials for land reclamation and recovery of degraded land. The evaluation includes the calorific value, carbon content, and bioaccumulation potential, with bioaccumulation potential envisioned for the phytoremediation of contaminated land [29]. Several species are integrated into these studies, namely *Cynara cardunculus* [30], *Populus euramericana* [31], *Miscanthus gigantheus*, and *Phalaris arundinacea* [32].

Sweet potato (*Ipomoea batatas* L.) is one of the most important vegetable crops in the archipelago of Madeira (Portugal), having been introduced to this region in the mid-17th century. In the warmer areas, plantations are carried out all year round, with several varieties existing along this archipelago, including the variety "Cabreira Branca" from Porto Santo [33]. The production area for this crop in the Madeira archipelago comprises 430 hectares, resulting in an annual production of 7351 tons in 2020 [34]. Due to such importance, this is the target crop of this study. The focus was to study the possibility of using sun-dried SS as a favorable fertilizer for sweet potato crops, to determine the heavy metal translocation between the soil and the different plant parts, and to determine whether the concentrations were below the imposed safe limits for food consumption. To our best knowledge, there are no studies comprising sweet potato agronomic assays with SS fertilization to determine heavy metal bioaccumulation. This will determine the feasibility of using this biomass generated locally in the production of a local variety of sweet potato in Porto Santo, giving a step forward to the biosustainability and circular economy of the Island.

## 2. Materials and Methods

### 2.1. Study Area

The assay was implemented in an experimental field with an area of 42.3 km$^2$ that is located on Porto Santo Island in the eastern Atlantic Ocean, about 40 km to the NE of Madeira Island [35]. Soils from Porto Santo reflect the dry climate to which they are naturally exposed. They are strongly degraded due to water and wind erosion, inadequate farming practices for cereal crops, and overgrazing [36]. The soil texture from the experimental field was classified as clay loam (28% sand, 38% silt, 34% clay), and it has a pH of 8.1 and 1.42% organic matter.

### 2.2. Agronomic Procedure

For this experiment, sweet potato (*Ipomoea batatas* var. Cabreira) was used as plant material. Dehydrated SS was used as a fertilizer, provided from a WWTP in Porto Santo. It comes exclusively from domestic effluents. The treatment system is based on an initial harrowing, a process of activated sludge in prolonged aeration, including the processes of denitrification/nitrification and secondary decantation. The resulting sludge is mechanically dewatered using a centrifuge after the addition of diluted polyelectrolyte and, finally, dried in a solar sludge drying oven (for details regarding physicochemical properties of dehydrated SS between 2009 and 2018 see Table S1).

Physicochemical properties of the receptor soil and the SS used in the experiment were determined previously according to Portuguese legislation (Table 1). The experiment occurred in two consecutive production cycles using a split-plot design. Two different sludge concentrations were tested per cycle, with three 25 m$^2$ plots each. For the first cycle, 20 kg and 40 kg of dehydrated sludge were mixed with soil to a depth of 0.2 m, obtaining a final concentration of 0.8 kg/m$^2$ (C1) and 1.6 kg/m$^2$ (C2), respectively. Half of the concentration was used for the second cycle (0.4 kg/m$^2$ and 0.8 kg/m$^2$). Three control plots with no sludge (C0) were used in each cycle. Fifty-eight sweet potato plants were planted in each plot, and five were then selected for individual analysis at the end of each cycle. A 2 m$^2$ area per replicate was delimited, and the plants in this area were collected for plot analysis. Agronomic parameters were measured according to Gança et al. (2018) and Anislag (2019) [37,38]. Parameters analyzed per plot were biomass (total fresh weight of shoot and tuberous roots), productivity (tuberous roots tons/hectare), shoot biomass, and harvest index (ratio between total tuberous root weight and total biomass, given as percentage). Parameters analyzed per plant were the number of tuberous roots/plant, tuberous root weight/plant, individual tuberous root weight, shoot weight/plant, shoot length/plant, and internode diameter/plant.

**Table 1.** Physicochemical properties of agricultural soil and sewage sludge that were used in the experimental assay.

| Properties | Agricultural Soil | Sewage Sludge | |
|---|---|---|---|
| | Measured Values | Measured Values | European Norms * |
| pH | 7.2 | 6.7 | NA |
| OM (%) | 1.42 | 74.1 | NA |
| CEC (meq/100 g) | 31.3 | ND | NA |
| NO$_3$-N (mg·kg$^{-1}$) | 3 | <0.5 | ** |
| NH$_4$-N (mg·kg$^{-1}$) | 2.3 | 6000 | ** |
| P (mg·kg$^{-1}$) | 1237 | 29,000 | ** |
| K (mg·kg$^{-1}$) | 960 | 10,000 | ** |
| Cd (mg·kg$^{-1}$) | ND | 0.7 | 20 |
| Cu (mg·kg$^{-1}$) | 10 | 110 | 1000 |
| Cr (mg·kg$^{-1}$) | ND | 28 | 1000 |

**Table 1.** *Cont.*

| Properties | Agricultural Soil | Sewage Sludge | |
|---|---|---|---|
| | Measured Values | Measured Values | European Norms * |
| Hg (mg·kg$^{-1}$) | ND | 0.3 | 16 |
| Ni (mg·kg$^{-1}$) | ND | 24 | 300 |
| Pb (mg·kg$^{-1}$) | ND | 21 | 750 |
| Zn (mg·kg$^{-1}$) | 6 | 690 | 2500 |

* Maximum permissible levels for heavy metals (mg·kg$^{-1}$) in sewage sludge to apply to agricultural soils according to DL nº 276/2009 (Portugal). ** It should consider the crop's needs and it cannot compromise superficial and underground water. ND—not determined. NA—not applicable.

### 2.3. Soil Physicochemical Properties

Soil samples were collected at the end of the cycles, air-dried, ground, and sieved (2 mm) for further analysis of pH, organic matter (OM), cation exchange capacity (CEC), and macronutrients (NO$_3$-N, NH$_4$-N, P, and K) at the Laboratory at the Directory of Laboratory and Agro-Food Research Services in Camacha, Madeira, Portugal. Ten grams of soil was suspended in 20 mL of potassium chloride 1 N for pH determination [39]. For OM, it followed the Walkley and Black method, with 0.5 g of soil digested with 10 mL of sodium dichromate 3 N solution in sulfuric acid 10 N [40]. Determination of CEC was done by percolation of 2 g of soil with 75 mL of ammonium acetate solution 1 M, at pH 7.0 [41]. Nitrate nitrogen and ammonium nitrogen were determined by continuous-flow auto analyzer after stirring 15 g of soil in 75 mL of bidistilled water. For potassium and phosphorus, we followed the Egner–Riehm method, with extraction of 2 g of soil with 40 mL of ammonium lactate solution in an acetic medium [42].

### 2.4. Heavy Metals

The concentration of the heavy metals cadmium (Cd), nickel (Ni), lead (Pb), mercury (Hg), and chromium (Cr) in the SS, soil, and plants were evaluated at the end of the assays. Triplicates were analyzed by a certified laboratory. Regarding SS and soil, the analysis of Cd, Ni, Pb, and Cr was done through inductively coupled plasma optical emission spectrometry (ICP-OES) following the ISO 11885:2007 [43], and Hg was analyzed through atomic absorption spectrometry (AAS) following the standard EN 1483:2007 [44]. For plants, the analysis of Cd, Ni, Pb, and Cr was done through ICP-OES following Hansen et al. (2013) [45], and Hg was analyzed through thermal decomposition, followed by a silver amalgamation, and quantified by ASS in a direct mercury analyzer (DMA-80, Milestones Srl, Sorisole, Italy). Data were used to compare treatments, compare with legislation, and determine the bioaccumulation factor (BF) and translocation factor (TF). The BF and TF were calculated according to Eid et al. [46] as follows:

$$BF = \frac{\text{Heavy metal content in the root } (mg \cdot kg^{-1})}{\text{Heavy metal content in the soil } (mg \cdot kg^{-1})}$$

$$TF = \frac{\text{Heavy metal content in the shoot } (mg \cdot kg^{-1})}{\text{Heavy metal content in the root } (mg \cdot kg^{-1})}$$

### 2.5. Statistical Analysis

Analysis of variance (ANOVA) and Tukey's HSD tests for agronomic parameters, soil properties, and heavy metal contents in soil and plants were performed using IBM SPSS for Windows, Version 23.0. The Pearson correlation between bioaccumulation and translocation factors and soil physicochemical properties was calculated in RStudio Version 1.3.1056, using the function ggcorr from GGally library, an extension of ggplot2.

## 3. Results

### 3.1. Agronomic Data per Plot

For the full experiment (first and second cycles together), the sweet potato biomass (total fresh weight of shoot and tuberous roots) obtained in an area of 2 m² of each plot was measured. We observed that the total biomass and shoot biomass is higher in C2 than in C0 and C1, but this difference is only statistically significant in C0 (Figure 1A,C), with C1 showing biomass greater than C0 but with no significant difference. Productivity was also higher in C2 than in C0 and C1, but this difference is only statistically significant for C1 (Figure 1B). C1 had lower productivity (tuberous roots) than C0 but higher biomass in the aerial part (not significant). The harvest index (ratio between the aerial component and tuberous roots) was higher in C0 and lower in C2, with significant differences between C0 and the two other treatments but not between C1 and C2 (Figure 1D). With the application of C2 for both cycles, a higher number for total biomass, productivity, and shoot biomass was observed, but this was not true for the harvest index.

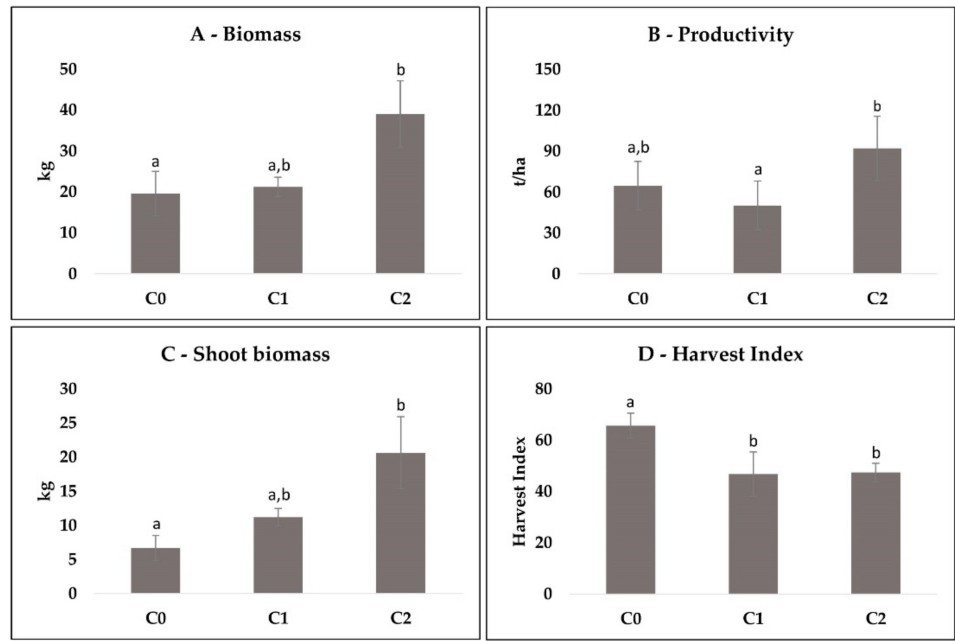

**Figure 1.** The figure indicates per-plot: (C0, C1, and C2 are the sewage sludge concentrations. Harvest index is indicated in percentage. Different letters indicate a significant difference among them ($p \leq 0.05$).

### 3.2. Comparison between Cycles per Plot

Comparing the two growth cycles, the total biomass increased in C0, increased significantly in C1, and remained practically the same in C2 (Figure 2A). Productivity increased slightly in C0, increased in C1, and decreased in C2, although not significantly (Figure 2B). The shoot biomass increased in all treatments but was statistically significant only in C1 (Figure 2C). The harvest index decreased in all treatments but only significantly in C2 (Figure 2D). In general, no significant difference was observed between cycles, except for C1 for biomass and shoot biomass and for C2 in the harvest index.

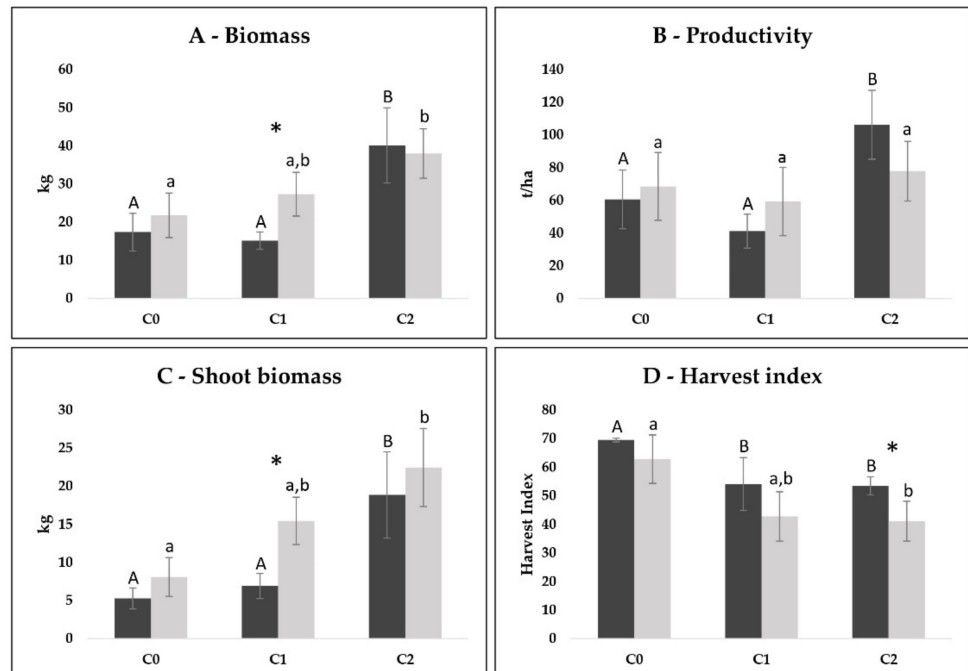

**Figure 2.** The figure indicates per-plot: C0, C1, and C2 are the sewage sludge concentrations. Harvest index is indicated in percentage. Bars identification: ■ = 1st cycle, ■ = 2nd cycle. Different letters indicate a significant difference among them ($p \leq 0.05$), within the cycle. Capital letters are for 1st cycle and lowercase letters are for 2nd cycle. The symbol * indicates a significant difference between growth cycles for sludge concentration ($p \leq 0.05$).

### 3.3. Agronomic Data per Plant

Analyzing the individual plants (five plants per plot), the same pattern was observed as the one for the plots, although with some statistical differences. The number of tuberous roots produced was higher in C2 and lower in C1, although not significantly (Figure 3A), with C0 being higher than C1 but lower than C2. Tuberous root biomass per individual plant was significantly higher in C2 compared to other treatments and lower in C1, although not significantly (Figure 3B), with C0 being higher than C1 but lower than C2. The weight of an individual tuberous root (Figure 3C) was significantly higher in C2 compared to the other treatments, followed by C0 and then C1, although the differences between these last two treatments are not statistically significant. The shoot weight per plant is significantly different between all treatments, being higher in C2, followed by C1, and having C0 producing significantly less shoot biomass than the other two treatments (Figure 3D). The shoot length per plant was significantly longer in C2 (Figure 3E) relative to both C1 and C0. Although C1 was slightly longer than C0, this is not significant. The internode diameter was slightly higher in C2, and this difference is significant for C0, which has the smallest thickness, but not for C1. C1 was thicker than C0, but the difference is not significant (Figure 3F). In half of the parameters (Figure 3A–C), the values follow the sequence C2 > C0 > C1, and for the other three parameters (Figure 3D–F), the same tendency was observed as in the data per plot (C2 > C1 > C0).

### 3.4. Comparison between Cycles per Plant

Comparing the two cycles, for the individual plants, a similar pattern to the one observed per plot is apparent, although with some differences. The number of tuberous roots produced was higher in the second cycle in C0 and C1 but lower in C2. This difference is only significant for C1 (Figure 4A). The biomass of the tuberous roots produced per plant was practically the same in the two cycles in C0, slightly higher in the second cycle in C1, and significantly lower in the second cycle in C2 (Figure 4B). The individual tuberous root weight was significantly lower in the second cycle in C0 and C2 and very similar in

the two cycles in C1 (Figure 4C). The shoot weight produced per plant was higher in the second cycle at all sludge concentrations, but this difference is only significant for C0 and C1 (Figure 4D). The length of shoots per plant was slightly higher in the second cycle in C0 and C1 and slightly lower in C2, although without significant differences (Figure 4E). The internode diameter was significantly smaller in the second cycle in all treatments (Figure 4F). Overall, a trend of lower values in the second cycle can be observed. Significant differences between cycles were detected for C2 in tuberous root weight, C0 and C1 for individual tuberous root weight, C0 and C1 for shoot weight, and for all treatments for the internode diameter.

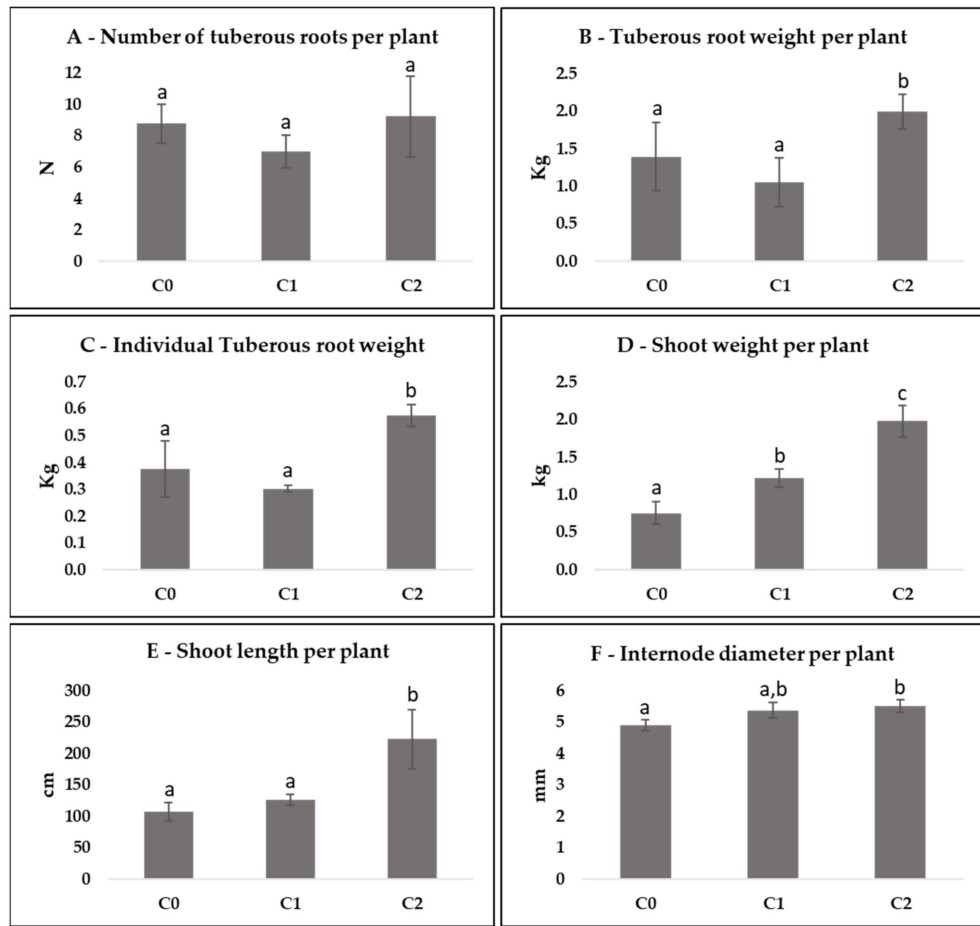

**Figure 3.** The figure indicates C0, C1, and C2 as the sewage sludge concentrations. Different letters indicate a significant difference among them ($p \leq 0.05$).

### 3.5. Soil Physicochemical Properties after SS Application

An ANOVA and a Tukey HSD test were applied to see if the differences were significant within the cycles and between treatments (Table 2). Organic matter (OM) showed a slight increase from C0 to C2 in the first cycle, although with no significant difference. The pH decreased in C2, with a significant difference in the first cycle. Cation-exchange capacity (CEC, based on Ca, K, Mg, and Na) was significantly higher in C2 of the second cycle. $NH_4$-N was improved in the first cycle, from C0 to C2, although with different behavior in the second cycle, where C1 showed the higher values. However, the best results were found for $NO_3$-N in both cycles, being significantly improved in C2.

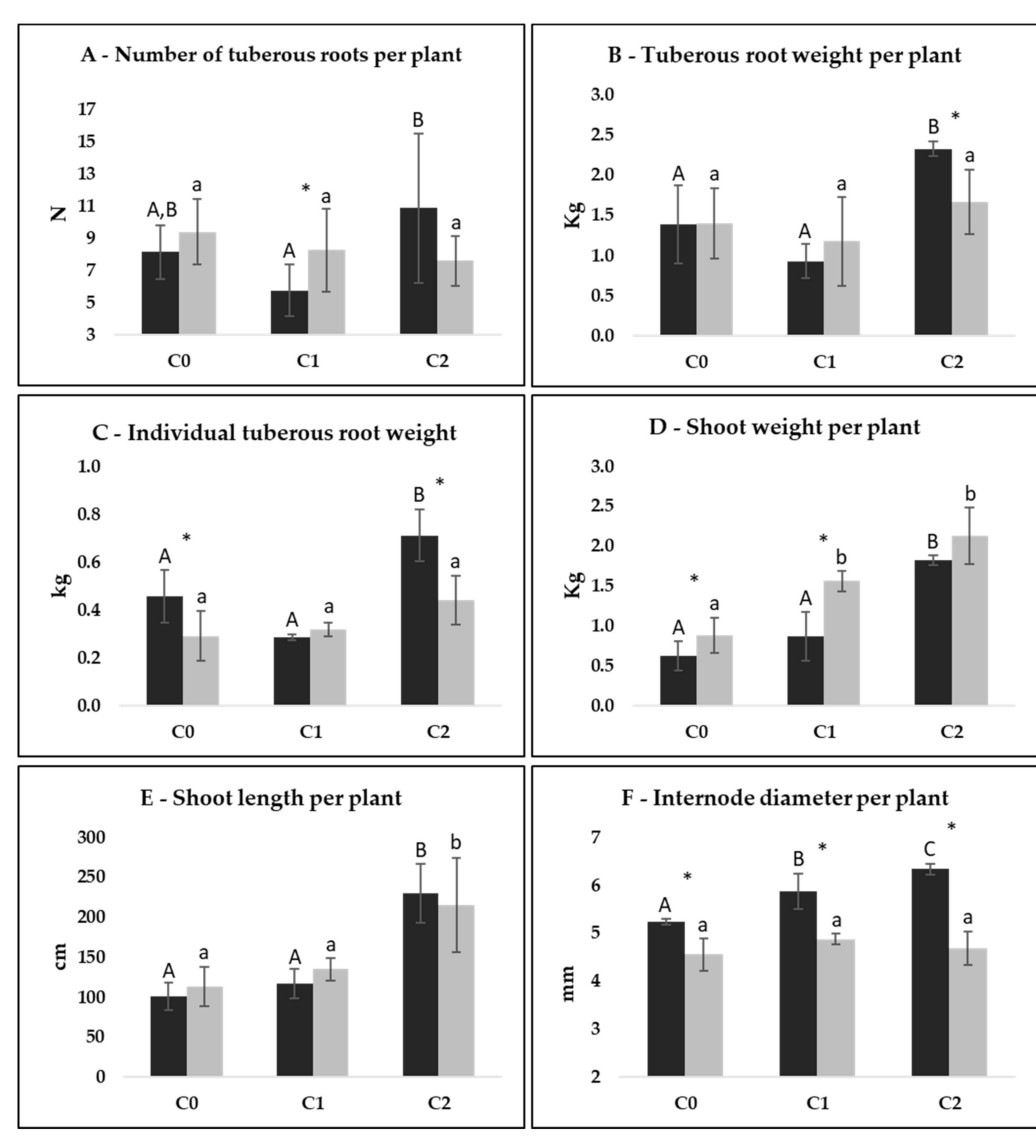

**Figure 4.** The figure indicates C0, C1, and C2 as the sewage sludge concentrations. Bars identification: ■ = 1st cycle ■ = 2nd cycle. Different letters indicate a significant difference between them ($p \leq 0.05$), within the cycle. Capital letters are for 1st cycle and lowercase letters are for 2nd cycle. The symbol * indicates a significant difference between growth cycles for that sludge concentration ($p \leq 0.05$).

**Table 2.** Physicochemical properties of postharvest soils at different SS concentrations (means ± standard error, $n$ = 3). Different letters next to the numbers indicate a significant difference between these values ($p \leq 0.05$), within the cycle. No letters mean no significant differences within the cycle for the given parameter.

| Properties | End of 1st Cycle | | | End of 2nd Cycle | | |
|---|---|---|---|---|---|---|
| | C0 | C1 | C2 | C0 | C1 | C2 |
| OM (%) | 1.48 ± 0.05 | 1.71 ± 0.16 | 1.75 ± 0.24 | 2.01 ± 0.00 | 1.91 ± 0.00 | 2.01 ± 0.00 |
| pH | 7.4 ± 0.05 a | 7.4 ± 0.05 a | 7.3 ± 0.05 b | 7.8 ± 0.00 | 7.8 ± 0.00 | 7.7 ± 0.00 |
| CEC (meq/100 g) | 33.7 ± 2.12 | 32.83 ± 3.24 | 34.57 ± 1.23 | 52.4 ± 0.00 a | 50.0 ± 0.00 a | 65.8 ± 0.00 b |
| $NO_3$-N (mg·kg$^{-1}$) | 22.5 ± 2.50 a | 27.5 ± 2.5 a | 105 ± 5.00 b | 15 ± 0.00 a | 35 ± 0.00 a.b | 55 ± 0.00 b |
| $NH_4$-N (mg·kg$^{-1}$) | 2.17 ± 0.17 a | 2.15 ± 0.15 a | 5.65 ± 0.05 b | 3.5 ± 0.00 a.b | 4.9 ± 0.00 a | 2.6 ± 0.00 b |
| P (mg·kg$^{-1}$) | 1374 ± 0.00 | 1374 ± 0.00 | 1374 ± 0.00 | 1511 ± 0.00 | 1511 ± 0.00 | 1511 ± 0.00 |
| K (mg·kg$^{-1}$) | 2240 ± 150 | 2080 ± 226 | 1880 ± 113 | 2400 ± 0.00 | 2520 ± 0.00 | 2160 ± 0.00 |

Values obtained for P and K were stable among treatments and cycles, revealing that the application of the SS amendment could not improve these parameters.

The overall results revealed some improvements in physicochemical properties at the end of both cycles in the plots amended with SS.

### 3.6. Effect of SS Application on Heavy Metals Contents

The different SS concentrations did not show significant differences regarding the content of the tested heavy metals (Cd, Cr, Hg, Ni, and Pb) in the post-harvested soils (Table 3). Values obtained for the three treatments in both cycles are below the maximum levels accepted for agricultural soils, according to the Portuguese legislation based on the Council Directive 86/278/EEC.

**Table 3.** Heavy metal contents of post-harvest soils at different SS concentrations (means ± standard error, $n = 3$). No significant differences were found among treatments for the given parameters.

| Heavy Metals | End of 1st Cycle | | | End of 2nd Cycle | | | European Norms * |
|---|---|---|---|---|---|---|---|
| | C0 | C1 | C2 | C0 | C1 | C2 | |
| | mg·kg$^{-1}$ | | | | | | |
| Cd | 0.7 ± 0.00 | 0.8 ± 0.00 | 0.7 ± 0.09 | 0.7 ± 0.00 | 0.8 ± 0.09 | 0.6 ± 0.05 | 4 |
| Cr | 143.3 ± 4.71 | 136.7 ± 4.71 | 146.7 ± 4.71 | 140 ± 0.00 | 136 ± 4.71 | 146.7 ± 4.71 | 300 |
| Hg | <0.1 | <0.1 | <0.1 | <0.1 | <0.1 | <0.1 | 2 |
| Ni | 88.0 ± 2.16 | 85.7 ± 1.89 | 90 ± 2.94 | 85.3 ± 3.68 | 81.0 ± 2.16 | 85.7 ± 3.30 | 110 |
| Pb | 14.3 ± 0.94 | 15.0 ± 0.82 | 14.0 ± 0.82 | 18.3 ± 2.05 | 15.3 ± 0.47 | 14.7 ± 1.25 | 450 |

* Maximum permissible levels of heavy metals in the soil after application of sewage sludge, considering pH > 7.0, according to Portugal's legislation (DL nº 276/2009).

Heavy metals were also analyzed in sweet potato tuberous roots and shoots (Figure 5). Overall results revealed lower contents of heavy metals in sweet potato tissues from both SS concentrations, except for Pb and Cr. The content of Pb was higher in C1 and C2 in tuberous roots and shoots in both cycles, although in the second cycle the differences are not significant. The highest Cr content, with a significant difference ($p \leq 0.001$), was found in tuberous roots from C1 in the first cycle, but no significant difference was found in the second cycle or for the shoot contents.

The content of heavy metals in tuberous roots and shoots were, in general, lower in the second cycle and there are no significant differences between concentrations.

The bioaccumulation factor (BF) and translocation factor (TF) values for the tested heavy metals are presented in Figure 6, showing the ability of sweet potato tissues to accumulate metals under SS concentrations. All the heavy metals showed values for BF lower than 1 in the three treatments. It was not possible to calculate the BF for Hg because the minimum value detected in the soil was 0.1 mg/kg (Table 3).

TF reached higher values than BF and in some cases was above 1. The TF for Cd was 3.14 in C2 (first cycle) but decreased from C0 (1.95) to C2 (0.83) for the second cycle. Hg shows an increase in TF value in the first cycle from C0 (0.78) to C2 (3.30), and a similar trend was observed for the second cycle, although C1 had the lower value, which was close to C0. Values for Cr were also close or above 1 in the second cycle: C1 (0.84) < C0 (0.98) < C2 (1.18).

The data in Figure 7 show that the BF of the tested heavy metals was negatively influenced mainly by OM (Cd > Ni > Pb > Cr). The pH is also correlated with BF for Pb ($r = 0.947$) and Cr ($r = 0.763$). On the other hand, TF seems to be influenced differently by heavy metals. Cr has a strong positive correlation with CEC ($r = 0.943$) and a less strong correlation with OM ($r = 0.780$) and pH ($r = 0.763$). Hg is positively correlated with OM ($r = 0.789$). Pb and Cd are linked mainly with the concentration of NH$_4$-N ($r = 0.718$; $r = 0.754$ respectively), and finally, Ni shows a strong negative correlation with pH ($r = 0.906$) and less strong correlation with CEC ($r = 0.749$), K ($r = 0.722$) and OM ($r = 0.684$).

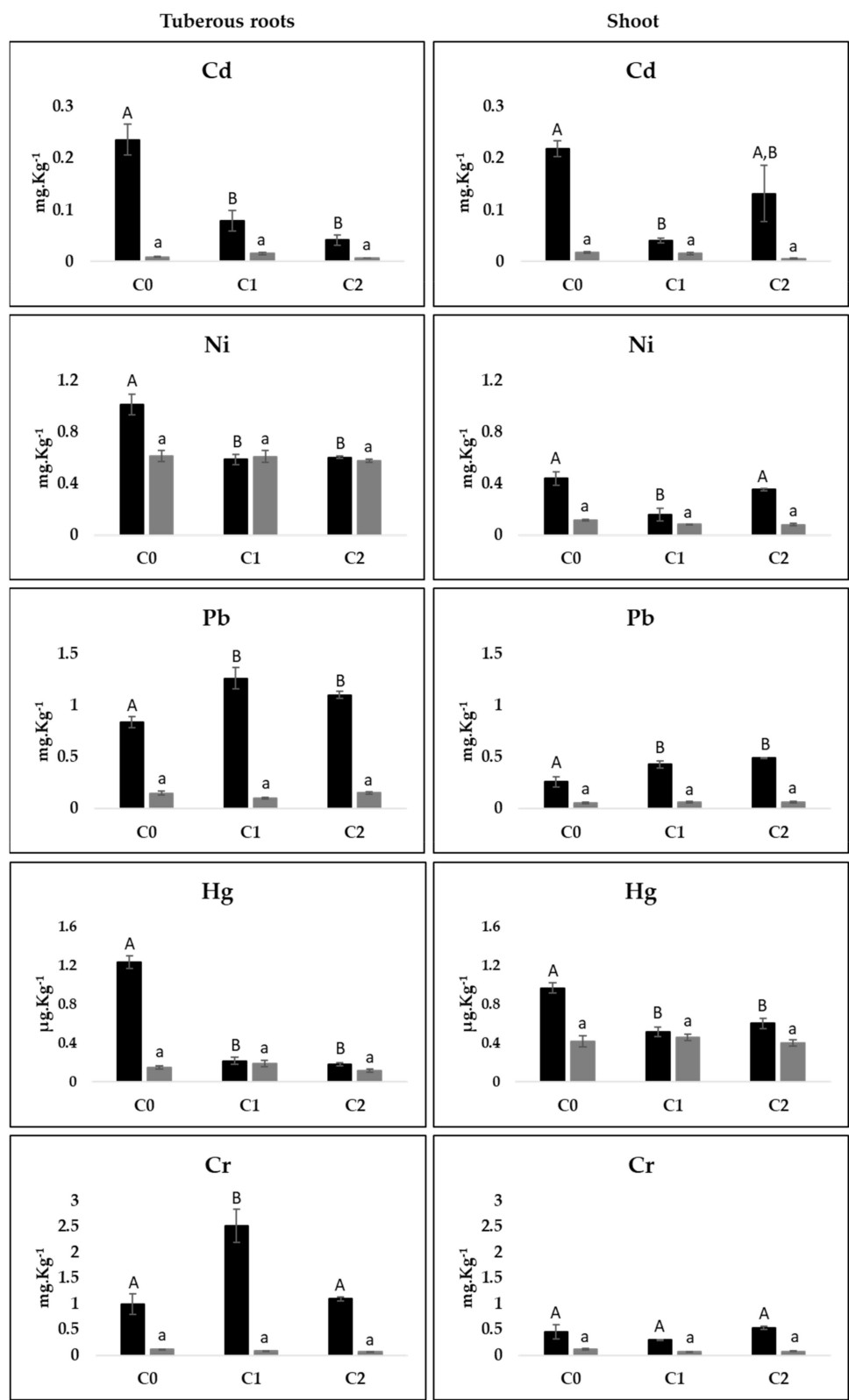

**Figure 5.** Heavy metal contents (mg·kg$^{-1}$) in tuberous roots (graphics on the left) and shoots (graphics on the right). Bars identification: ■ = 1st cycle ■ = 2nd cycle. C0, C1, and C2 are the sewage sludge concentrations. Bars with different letters indicate a significant difference ($p \leq 0.05$) between them, within the cycle. No letters mean no differences within the cycle. Capital letters are for 1st cycle and lowercase letters are for 2nd cycle.

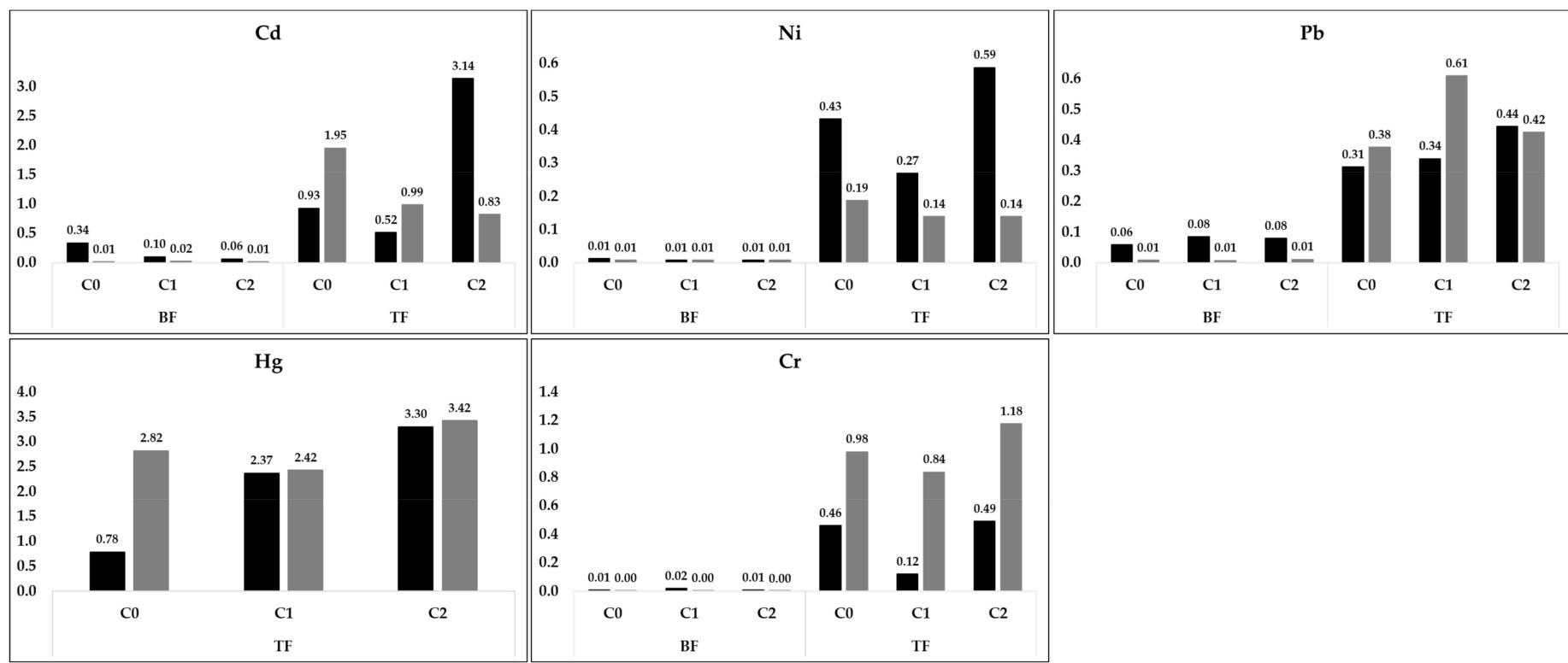

**Figure 6.** Bioaccumulation factors (BF) from soil to roots and translocation factors (TF) from roots to shoots for heavy metals in sweet potato grown in soil amended with different SS concentrations. Bars identification: ■ = 1st cycle ■ = 2nd cycle. C0, C1, and C2 are the sewage sludge concentrations.

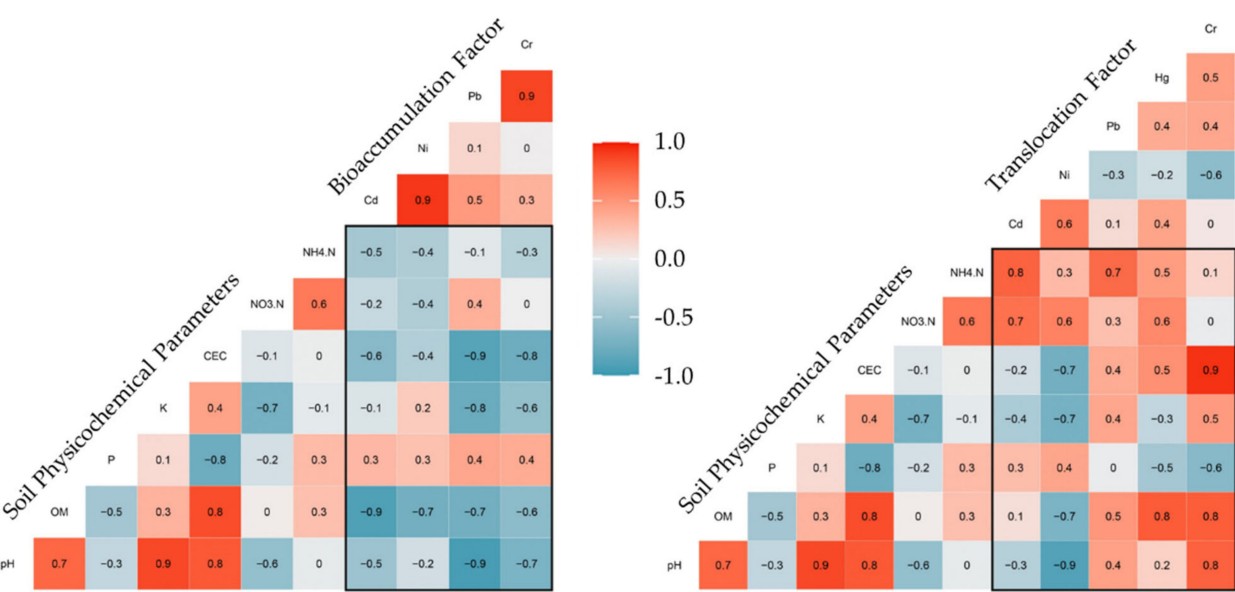

**Figure 7.** Correlation matrix between bioaccumulation factor (BF) and translocation factor (TF) for heavy metals and soil physicochemical parameters.

All the heavy metals for the three treatments showed values lower than 1 for BF (except Hg), but Cd, Hg, and Cr presented values higher than 1 for TF.

## 4. Discussion

### 4.1. Agronomic Data per Plot

The role of SS in enhancing the productivity parameters has been well demonstrated in several studies (e.g., [24,46–49]). In this work, we aimed to evaluate the use of SS as a source of organic matter for the re-generation of eroded soils and the improvement of agricultural production. Our results showed that SS application improved the agronomic parameters of biomass, productivity, and shoot biomass but had the opposite effect on the harvest index. This tendency was observed per plot when the cycles were analyzed together and compared between cycles. Our results are corroborated with other works regarding improving agronomic parameters using SS. For example, regarding sweet pepper plants, a study in Spain monitored the effect of the application of three increasing amounts of SS (3, 6, and 9 kg·m$^{-2}$) on the physicochemical properties of calcareous soil. Pepper fruit biomass production in the greenhouse was almost 60% higher than that of the open-air plot, with a dose of 6 kg·m$^{-2}$ [14]. In another study, a 3-year experiment assessed the SS application and its residual effect on wheat crop productivity. The results demonstrated that a dose of 2.5 kg·m$^{-2}$ per SS application provides the highest crop productivity [50]. For maize, a study evaluated two forms of SS stabilization (chemically stabilized and composted) on the shoot biomass, yield, and concentration of nutrients in the soil and plants in three successive cycles. In the first harvest, maize yield using the chemically stabilized SS was 34% higher than that obtained with mineral fertilization [47]. In the case of barley, based on the results of a study with different SS concentrations, the recommended dose for the best growth of barley plants was 40 g SS per kg of soil [51]. In a study performed in Saudi Arabia, it was concluded that the SS concentration of 20 g·kg$^{-1}$ triggered the highest growth rates forf *Corchorus olitorius*, a commonly consumed leafy vegetable crop [52]. In a pot experiment, Kumar and Chopra detected the maximum growth of *Phaseolus vulgaris* (shoot/root length, biomass, and crop yield) in the treatment with 40% sludge + 60% garden soil [22]. In a study from India with mung beans, the authors suggest that SS at a rate lower than 9 kg·m$^{-2}$ may be recommended due to the promotion of the mung bean yield [23]. Although rare in the bibliography, examples regarding the detriment of power can be found. A study of palak (*Beta vulgaris* var. Allgreen) observed that SS amendment in soil decreased the root

length, leaf area, and root biomass of palak at both ratios (20% and 40%), whereas shoot biomass and yield decreased significantly at 40% [25].

The percentage of the harvest index (ratio between total tuberous root weight and total biomass) was higher in C0 than in C1 and C2, both when the cycles were analyzed together or in the comparison between cycles. The index followed the opposite tendency of the other agronomic parameters and indicates that the plant invested more energy in vegetative growth rather than that of the tuberous root (for further detail see point 4.3).

### 4.2. Agronomic Data per Plant

The parameters per plant (no. of tuberous roots, tuberous root weight, individual tuberous root weight, shoot weight, shoot length, and internode diameter) follow the same pattern observed in the analysis of the plot, presenting the higher values for the higher SS concentration (C2). Agronomic parameters analyzed per plant are important to assess the necessary characteristics for the product normalization/caliber and the overall market quality. For example, quality parameters such as size, weight, shape, absence of defects, and nutritional composition are important when considering the tuberous root acceptance by the market [53].

One of the main factors influencing sweet potato productivity and quality is the supply and soil availability of both macronutrients and micronutrients [54], and in this study the application of SS improved the availability of those nutrients, thus improving agronomic parameters.

The agronomic parameter improvement, in addition to the previous implications such as tuber yield, is also important when considering cattle feed and soil mulching, with the improvement of shoot weight and shot length.

### 4.3. Impact on Soil Physicochemical Properties

The post-harvest analysis of the soil's physicochemical properties revealed some positive changes in the plots amended with SS compared to C0 (control), mainly for pH, CEC, $NO_3$-N, and $NH_4$-N. Plots with the highest concentration of SS (C2) showed the best results, and this is in line with the agronomic outcomes. The productivity was higher in C2; however, the harvest index was lower than C1 and C0, meaning that the plant invested in vegetative growth. The literature reports the positive effect of nitrogen on vegetative growth and its direct impact on the final crop yield [55–58]. Indeed, there was a significant improvement in $NO_3$-N concentration in plots with the highest concentration of SS. Binder et al. [59] also showed benefits in the form of N due to the application of SS.

It is expected that the application of SS increases OM content significantly [4,46,60]. Although in this study there was a slight increase in this property in the first cycle, this was not statistically significant. Probably, OM from SS was quickly biodegraded, which led to a slight decrease in pH and a release of nutrients, including $NH_4$-N [60,61].

Contrary to the other improved properties, CEC had a better and significant improvement in the second cycle. Other studies reported this improvement after years of applications for SS [62–64]. This increment, associated with the high pH (>7.0) of the studied soil, may lead to a reduction in heavy metal mobility [64].

### 4.4. Heavy Metal Content in Soil and Sweet Potato Tissues

The application of SS on agricultural soils may lead to the accumulation of heavy metals that can eventually be absorbed by the plants, reaching toxic levels and entering the food chain [52]. This concern led the Council of the European Communities to regulate the use of SS in agriculture to prevent harmful effects to soil, plants, animals, and humans [18]. In this study, the heavy metal content in the soil did not show significant differences among treatments, and all the values were below the maximum permissible levels in the soil after the application of SS. These results show that the concentrations used in this soil were appropriate. Nevertheless, it is important to consider that this study was conducted for

2 years, with the concentration in the second cycle being half in comparison with the first cycle. Long-term applications can lead to different results [64].

The accumulation of heavy metals in plants depends on SS contents and concentration but also the properties of the soil, the metals' nature, and the plant species [62,65]. In the second cycle, the pH was near 8.0 in all the experimental plots, and CEC increased in the highest SS concentration (C2). These abiotic conditions probably had an impact on heavy metal mobility, including for Pb and Cd, which were above permissible levels in the first cycle in tuberous roots and leaves, according to Commission Regulation (EC) Nº 1881/2006 [20]. The maximum permissible levels are 0.1 mg·kg$^{-1}$ (wet weight) for both metals in root vegetables. Regarding leaves, the maximum levels are 0.1 for Pb and 0.2 mg·kg$^{-1}$ (wet weight) for Cd. However, Pb remained at unsafe values for all the samples, including the control, in tuberous roots. The application of SS can have a positive impact on this problem in the long term, improving soil properties, mainly CEC and OM, as Pb tends to bind strongly to organic and colloidal materials and is thereby less available for plant uptake [66]. In addition, in this study, the increase of SS concentration led to a decrease in Cd content in tuberous roots, and there was also a significant decrease in Ni and Hg contents in samples from plots with SS when compared to the control (without SS). On the other hand, the content of SS and the accumulation of these and other metals in soils and plants should be monitored.

The heavy metals Ni and Pb were accumulated in higher amounts in tuberous roots, while Cd, Hg, and Cr adopted different behaviors among treatments and cycles. The ability of plants to absorb and translocate heavy metals from roots to shoots is determined by calculating the bioaccumulation factor (BF) and translocation factor (TF). These factors may change as the soil properties change [66]. The literature reported that plants with BF and/or TF values above one are considered hyperaccumulators [46,66]. According to our data, sweet potato was revealed to be a hyperaccumulator of Cd, Hg, and Cr in shoots, under certain conditions. The TF of Cd seems to be influenced by the N concentration, mainly NH$_4$-N. Indeed, several studies show the effect of N in Cd uptake by plants [67–70]. NO$_3$-N and NH$_4$-N differentially alter Cd uptake, accumulation, and chemical speciation in plants, and Cheng et al. [70] showed that Cd translocation to shoots in wheat is promoted by NH$_4$-N. This is probably why there was a higher TF in C2 in the first cycle.

Some studies reported that Hg is preferably accumulated in roots instead of shoots [71,72]; however, in this study, this is only true in C0 for the first cycle. The Pearson correlation indicates OM as a possible regulator of Hg translocation. Muddarisna and Siahaan [73] found higher values for TF when organic matter was added to the medium, suggesting that this addition changes the role of the plants under study from phytostabilization to bioaccumulation, which translocate Hg from the roots to the shoots. The same seems to have happened regarding Cr, but in this case, the soil parameter that may have been affected was CEC. Compared to other heavy metals, the mobility of Cr within the plant is limited and depends, mainly, on the chemical form of Cr [74]. This limited mobility was observed in this study during the first cycle. However, in the second cycle, although the contents were low, the TF in C2 exceeded 1. This change in the translocation ability of Cr should be observed and better studied, as this metal impairs plant photosynthetic efficiency [74].

## 5. Conclusions

The extensive range of biochemical compounds dissolved in the wastewater originates from households and industrial processes and contributes directly to the SS composition and, consequently, its use for crop fertilization. The results from this research study demonstrated that SS application to low-fertility soil is effective in improving crop agronomic parameters in sweet potato. In addition, it was possible to observe improvements in soil properties, mainly for pH, CEC, NO$_3$-N, and NH$_4$-N, with no significant increase in soil heavy metals. Nevertheless, some heavy metals exceeded maximum permissible limits in the edible parts of the plant in both cycles, namely Pb. The existing excess of Pb in the

samples from control plots and the drastic decrease in the second cycle led us to question whether the improvements in soil properties observed with the application of SS could have a positive impact on this issue when soils have pH close to 8.

These results indicate the possibility to use the investigated SS in agriculture according to the European directives. On the other hand, it is difficult to extrapolate the results from short-term experiments (two cycles) to assess the potential impact in the long term. As far as we know, this is the first study comprising sweet potato agronomic assays with SS fertilization to determine heavy metal bioaccumulation, and studies like this can help to understand the beneficial/harmful effects of SS application and generate data that contribute to European guidelines. Notwithstanding this, we consider that further studies taking into consideration these variables, namely the SS origin, soil physicochemical properties, and the crop must be carried out to propose adequate applications.

**Supplementary Materials:** The following supporting information can be downloaded at: https://www.mdpi.com/article/10.3390/agronomy12081902/s1. Table S1: Physicochemical properties, including heavy metals, of dehydrated sewage sludge from the wastewater refinement facility of Porto Santo between 2009 and 2018.

**Author Contributions:** Conceptualization, C.R., N.N., M.C.O.O. and J.G.R.d.F.; investigation, C.R., N.N., M.C.O.O. and J.G.R.d.F.; writing—original draft preparation, C.R., N.N. and M.C.O.O.; writing—review and editing, C.R., N.N. and J.F.T.G.; supervision, M.Â.A.P.d.C.; funding acquisition, N.N., M.C.O.O., J.F.T.G. and M.Â.A.P.d.C. All authors have read and agreed to the published version of the manuscript.

**Funding:** This research was funded by *Programa Operacional Madeira* 14–20, Portugal 2020, and the European Union through the European Regional Development Fund, grant number M1420-01-0145-FEDER-000011 [CASBio] and by the Agência Regional para o Desenvolvimento da Investigação, Tecnologia e Inovação, Portugal 2020 and the European Union through the European Social Fund [grant number M1420-09-5369-FSE000002, ARDITI].

**Acknowledgments:** The authors acknowledge the support by National Funds FCT-Portuguese Foundation for Science and Technology under the projects UIDB/04033/2020 and UIDP/04033/2020. Also, to the *ARM—Águas e Resíduos da Madeira, S.A.* for providing support and the sewage sludge biomass.

**Conflicts of Interest:** The authors declare no conflict of interest.

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
