# Peer review of "Sewage Sludge Fertilization—A Case Study of Sweet Potato Yield and Heavy Metal Accumulation"

_agronomy, doi:10.3390/agronomy12081902_

Round 1

Reviewer 1 Report

Many works were published in this field and I was surprised that in the introduction few works were cited concerning the aerobic and anaerobic digestion of SS. In fact the introduction should be improved:

- please remove the process to obtain SS from the introduction to the material and methods.

- give the actual situation of use of SS in Portugal in general and in your region in particular.

- give more info about the actual legislation of using SS in agriculture

- what about the regions producing more than 265 t of SS

Material and methods

give small description of analysis methods

Why total N was not measured in the soil?

Table 1: please verify your values.   For example K in SS 10 000 g/kg that means 10 kg/kg ???!

In this table instead of "Max levels" i prefer European Norms 

Results 

I am really surprised but the values of harvest index.

I am also surprised by the concentrations of heavy metals in the tubercule for C0. Why they are so high?

OM did not increased enough. 

Why you did not choose to compost the SS with green waste for example before incorporation in soil? The results could be better.

Author Response

Response to Reviewer 1 Comments

 Manuscript ID: agronomy-1846146

Type of manuscript: original article

Title: The effect of sewage sludge fertilization on the agronomic features of sweet potato

Authors: Carla Ragonezi, Nuno Nunes, Cristina Oliveira, José G. R. Freitas, José Filipe T. Ganança, Miguel Â. A. Pinheiro de Carvalho.

The authors appreciate the reviewer’s comments about the manuscript. The comments were addressed properly below and can be visualized in the resubmitted version.

Point 1: Many works were published in this field and I was surprised that in the introduction few works were cited concerning the aerobic and anaerobic digestion of SS. In fact, the introduction should be improved:

- please remove the process to obtain SS from the introduction to the material and methods.

- give the actual situation of use of SS in Portugal in general and in your region in particular.

- give more info about the actual legislation of using SS in agriculture

- what about the regions producing more than 265 t of SS

Response 1: The authors appreciate the reviewer´s comment and proceed to the correction in the introduction (please see the revised version).

Several works describing the aerobic and anaerobic digestion of SS were integrated and presented from line 48.

The process to obtain the SS was removed from the introduction and added to the methodology (see from line 121).

The SS´s use in Portugal and Madeira is added from line 66.

More information about the actual legislation for agriculture was added from line 78.

The authors inform that regarding the “regions producing more than 265 t of SS” this is the only WWTP implanted on the island. This was better described from line 75.

Point 2: Material and methods:

- give small description of analysis methods

- Why total N was not measured in the soil?

- Table 1: please verify your values. For example K in SS 10 000 g/kg that means 10 kg/kg ???!

- In this table instead of "Max levels" i prefer European Norms 

Response 2: The authors appreciate the reviewer´s comment and proceed to the correct and detailed information in the methodology (please see from line 150, Table1 and 4, and Supplementary Table 1).

Regarding the question about total N, it was not possible, at the time, to measure this parameter in soil samples. So, we thank the reviewer to point that out and we decided to remove this parameter from the tables, as this is not discussed further.

Point 3: Results 

I am really surprised but the values of harvest index.

I am also surprised by the concentrations of heavy metals in the tubercule for C0. Why they are so high?

OM did not increased enough. 

Why you did not choose to compost the SS with green waste for example before incorporation in soil? The results could be better.

Response 3: The authors appreciate the reviewer´s comment.

Regarding the high concentration of heavy metals in the tubercule for C0, one hypothesis is that the soil without the SS addition did not present high values of CEC, allowing greater availability of these metals in the soil and possibly allowing the absorption by the plants. In this case, the potato root is the energy storage organ, so it is more likely to find the greatest amount of heavy metals there.

Concerning organic matter, probably, the OM from SS was quickly biodegraded, since it was easily available to be used, which led to a slight decrease in pH and a release of nutrients, including NH4-N. Also, the plants could use the OM to be incorporated, contributing to the data regarding the agronomic parameters, for example, biomass. This information is also in the discussion section (line 396).

The choice to use the SS without composting is because parallelly we are performing an experiment with the SS composted. Once the data is analyzed, we can compare the results.

Reviewer 2 Report

Dear authors, I appreciate your well written manuscript on “The effect of sewage sludge fertilization on the agronomic features of sweet potato”. The manuscript is well organized, the topic of research is good. It's an important work, and the manuscript has been well prepared. I highly recommended to publish this manuscript in “Agronomy”, after major revision. Below is my assessment on the of the submitted manuscript. Based on the these; the manuscript can be accepted for publication after the major revisions are addressed.

Comments:

Title

The author should rework on the title of the manuscript to make it more attractive and accurately describe the content.

Abstract

Please focus on the novelty of this work.

The presentation of the key findings of experimental results should be improved and data regarding the mainly measured indicators should be presented.

Please consider changing the keywords list and use synonyms.

Introduction

Line 36: Do not begin the sentence with an abbreviation.

The present introduction describes what the authors hoped to achieve. However, the literature is quite repetitive in certain parts of the introduction. Thus, literature should be concentrated explaining how this study should add any significant improvement to the common knowledge.

The authors are recommended to add more details on the significance of the study and provide a hypothesis of the present study at the end of the introduction to give the reader more information regarding the purpose and the mechanistic used to achieve this goal, then may refer some lack in the previous study regarding some aspects.

Materials and Methods

The methodology is described insufficiently, this section is missing some details

How did the authors measure heavy metals ??? add a reference https://doi.org/10.1016/j.chemosphere.2022.134044

How did the authors measure plant morphological parameters ??? add a reference https://doi.org/10.1007/s42729-021-00727-2

Results

The results are sufficiently presented and readable, in particular statistical results, however this section lacks state of art and need one sentence at the end of each paragraph to show to readers what happen in the whole paragraph.

Discussion

The discussion part is well presented and well interrupted.

Conclusion

Well written

Kind Regards,

Author Response

Response to Reviewer 2 Comments

Manuscript ID: agronomy-1846146

Type of manuscript: original article

Title: The effect of sewage sludge fertilization on the agronomic features of sweet potato

Authors: Carla Ragonezi, Nuno Nunes, Cristina Oliveira, José G. R. Freitas, José Filipe T. Ganança, Miguel Â. A. Pinheiro de Carvalho.

The authors appreciate the reviewer’s comments about the manuscript. The comments were addressed properly below and can be visualized in the resubmitted version.

Point 1: Dear authors, I appreciate your well written manuscript on “The effect of sewage sludge fertilization on the agronomic features of sweet potato”. The manuscript is well organized, the topic of research is good. It's an important work, and the manuscript has been well prepared. I highly recommended to publish this manuscript in “Agronomy”, after major revision. Below is my assessment on the of the submitted manuscript. Based on the these; the manuscript can be accepted for publication after the major revisions are addressed.

Response 1: The authors appreciate the reviewer´s comments and overall observations.

Point 2: Title - The author should rework the title of the manuscript to make it more attractive and accurately describe the content.

Response 2: The authors appreciate the reviewer´s comment and proceed to the alteration of the title (please see the revised version).

Point 3: Abstract

- Please focus on the novelty of this work.

- The presentation of the key findings of experimental results should be improved and data regarding the mainly measured indicators should be presented.

- Please consider changing the keywords list and use synonyms.

Response 3: The authors appreciate the reviewer´s comment and proceed to the alteration in the abstract (please see the revised version).

Suggestions were accepted but not a lot of information and details could be changed since we have a limitation of 200 words in the abstract section.

Keywords were changed or deleted.

Point 4: Introduction

Line 36: Do not begin the sentence with an abbreviation.

The present introduction describes what the authors hoped to achieve. However, the literature is quite repetitive in certain parts of the introduction. Thus, literature should be concentrated explaining how this study should add any significant improvement to the common knowledge.

The authors are recommended to add more details on the significance of the study and provide a hypothesis of the present study at the end of the introduction to give the reader more information regarding the purpose and the mechanistic used to achieve this goal, then may refer some lack in the previous study regarding some aspects.

Response 4: The authors appreciate the reviewer´s comment and proceed to the alteration in the introduction (please see the revised version).

Line 36 was altered.

The introduction was significantly modified, and more details were added, explaining the importance of this study.

A hypothesis was described and located at the end of the introduction and also referred to the lack of research in this area, justifying this study.

Point 5: Materials and Methods

The methodology is described insufficiently, this section is missing some details

How did the authors measure heavy metals ??? add a reference https://doi.org/10.1016/j.chemosphere.2022.134044

How did the authors measure plant morphological parameters ??? add a reference https://doi.org/10.1007/s42729-021-00727-2

Response 5: The authors appreciate the reviewer´s comment and proceed to the correct and detailed information in the methodology (please see from line 150).

Point 6: Results

The results are sufficiently presented and readable, in particular statistical results, however this section lacks state of art and need one sentence at the end of each paragraph to show to readers what happen in the whole paragraph.

Response 6: The authors appreciate the reviewer´s comment and proceed to the correction in the results (please see the revised version).

The final sentence in each paragraph was added, but reasonably so that there was no repetition in the discussion.

Regarding the lack of state of art, since we did the results separated from the discussion, the state of art is on the latest one.

Point 7: Discussion

The discussion part is well presented and well interrupted.

Response 7: The authors appreciate the reviewer´s comment.

Point 8: Conclusion

Well written

Response 8: The authors appreciate the reviewer´s comment.
